# DFT Insight into Conductive and Magnetic Properties of Heterostructures with BaTiO_3_ Overlayer

**DOI:** 10.3390/ma15238334

**Published:** 2022-11-23

**Authors:** Alina Zagidullina, Irina Piyanzina, Zvonko Jagličić, Viktor Kabanov, Rinat Mamin

**Affiliations:** 1FRC Kazan Scientific Center of RAS, Zavoisky Physical-Technical Institute, 420029 Kazan, Russia; 2Institute of Physics, Kazan Federal University, 420008 Kazan, Russia; 3Institute of Mathematics, Physics and Mechanics, Jadranska 19, 1000 Ljubljana, Slovenia; 4Faculty of Civil and Geodetic Engineering, University of Ljubljana, Jamova c. 2, 1000 Ljubljana, Slovenia; 5Department for Complex Matter, Jozef Stefan Institute, 1000 Ljubljana, Slovenia

**Keywords:** ferroelectric, heterostructure, density functional theory, 2DEG, silicon

## Abstract

The ab initio calculations of a heterostructure based on the ferroelectric phase of barium titanate and dielectrics lanthanum manganese (LaMnO_3_) or silicon (Si) are presented. We analyze structures of BaTiO_3_/LaMnO_3_ and BaTiO_3_/Si interfaces, investigate magnetic properties and the impact of ferroelectric polarization. The use of ferroelectrics in the heterostructure plays a crucial role; in particular, ferroelectric polarization leads to the appearance of the conducting state at the interface and in the layers close to it. We show that defects (here, oxygen vacancies) incorporated into the system may change the electronic and magnetic properties of a system. Experimental results of magnetic susceptibility measurements for the Ba_0.8_Sr_0.2_TiO_3_/LaMnO_3_ heterostructure are also presented. It is shown that a correlation between the behavior of the ferromagnetic ordering and the resistance takes place. In addition, the ferromagnetic ordering at the interface of the heterostructure can be associated with the exchange interaction through current carriers that appear in high carrier concentration regions.

## 1. Introduction

The extensive investigation of heterostructures began in 2004 with the discovery of a two-dimensional electron gas (2DEG) [1]. The main interest to that branch of condensed matter physics is associated with the variety of physical phenomena observed at the interface of two or more components. Moreover, it became possible to combine incompatible properties in one material, for example, superconductivity [2,3,4] and magnetism [5], at the interface. The heterostructures from previous works on 2DEG were primarily LaAlO_3_/SrTiO_3_ (LAO/STO). There, the interfacial magnetism is very weak due to the non-magnetic nature of the starting material. It was also shown that magnetism in such systems arises due to the presence of defects [6,7,8,9,10,11]. For example, the maximum magnetic moment created by an oxygen vacancy located at the LAO/STO interface is 0.2 μ_B_ per Ti atom located in the vicinity of a defect [12]. Despite the great interest in heterostructures over the past two decades and a substantial amount of research, this area is still fascinating and attracts attention with the discovery of new phenomena [13,14,15,16,17]. To enhance the interfacial magnetism, it is necessary to use magnetic insulators as part of heterostructures to create a spin-polarized 2DEG. Another possibility is to have defects in the structure, which is not a complicated task since defects are ubiquitous in most experiments with oxide heterostructures growth. Even more promising is the manipulation of conducting states by an external electric field using ferroelectric materials [13,14,15,16,17], as well as the implementation of the inverse magnetoelectric effect, when the magnetization can be changed by an electric field [16].

The motivation for the BaTiO_3_/Si heterostructure was provided by the fact that silicon is still the main component of most microelectronic devices and a subject of scientific investigations [18,19]. In addition, BaTiO_3_ is a promising material for electro-optical (EO) modulators due to its large effective Pockels coefficient, especially in epitaxial form. Recently, the electrooptical properties of BaTiO_3_ epitaxial films on SrTiO_3_/Si were reported for the first time [20]. It has been shown that BaTiO_3_ exhibits a much higher effective Pockels coefficient, at least five times larger than LiNbO_3_. Being a ferroelectric, BTO has a unique crystallographic direction in which the ferroelectric polarization is indicated. It is also proposed to use BaTiO_3_ integrated in Si as a modulator for silicon photonics since silicon itself has a high energy consumption [21]. In addition, there is a huge variety of other applications of heterostructures based on BaTiO_3_: among others, plasmonic photocatalysts and piezocatalysis applications [22,23,24], as a device platform for ethanol vapor detection [25], data storage application as well as tunable functionalities such as memristor behavior [26]. The ferroelectric properties of BTO-based heterostructures reignited the interest in the development and implementation of alternative ferroelectric memory concepts, such as ferroelectric field-effect transistors (FFET) [27,28,29,30] and negative-capacitance field effect transistors (NCFET) [31,32]. To date, the FFETs have been demonstrated to exhibit either good retention or endurance, whereas complementary performance characteristics were poor. It is shown that the range of characteristics is quite wide and varies from a few second/hour retention time and endurance up to 10^12^ cycles to a ten-year retention time and endurance down to ~10^3^–10^4^ cycles [30].

The aim of the present study is to provide a comparative study of the electronic and magnetic properties of two different structures, with BaTiO_3_ as an overlayer used as a tool for the property’s manipulation. The essential aim is to uncover the common features of heterostructures with ferroelectric overlayer and to evaluate the role of BaTiO_3_. To do so, we used the density functional theory (DFT) as a comprehensive tool for the prediction of the electronic and magnetic properties. We present here computation results for BaTiO_3_/Si and for BaTiO_3_/LaMnO_3_ heterostructures complemented by the magnetic susceptibility measurements for the Ba_0.8_Sr_0.2_TiO_3_/LaMnO_3_ heterostructure.

## 2. Materials and Methods

### 2.1. DFT Details

This research contains ab initio calculations based on the density functional theory [33,34]. The exchange and correlation effects were accounted for using the generalized gradient approximation (GGA-PBE) [35]. The Kohn-Sham equations were solved using the plane wave basis, implemented in the VASP software [36,37,38] as a part of the MedeA computational environment [39]. The cut-off energy of 400 eV was used, the residual force was 0.05 eV/Å, and the total energy was equal to 10^−5^ eV. The Brillouin zone was split using a 5 × 5 × 1 grid. The calculations were completed using the simplified GGA + *U* method, proposed by S.L. Dudarev [40]. We have chosen the following parameters: +*U* = 2 eV, 4 eV, and 8 eV [41,42] for the Ti 3*d*, Mn 3*d,* and La 4*f* orbitals during the optimization and calculation of electronic and magnetic properties.

The heterostructures supercells modeling was performed by adding the BaTiO_3_ overlayers with interfacial TiO_2_ layers toward the substrate and BaO surface layers. In addition, the overlayers were added on both sides of the substrate, symmetrically, with respect to the center of the Si substrate or LaMnO_3_ slab. The underlying model guarantees a heterostructure without any artificial dipoles. In addition, a vacuum region was added to prevent unwanted interaction between surfaces and their periodic copies.

### 2.2. Resistivity and Magnetic Susceptibility Measurements Details

The electrical and magnetic properties have been studied for heterostructures formed by antiferromagnetic LaMnO_3_ single crystals with epitaxial films of ferroelectric Ba_0.8_Sr_0.2_TiO_3_ (BSTO) were deposited on top of them by reactive sputtering of stoichiometric targets, using the RF-sputtering method [42,43] at 650 °C. The average roughness of the LaMnO_3_ single crystals surface was approximately 1 nm before deposition. The results are presented for the sample in the which the thickness of the BSTO film was 453 nm. The BSTO film was in the tetragonal phase and in the ferroelectric state, as will be shown below. The electrical resistance was measured at the DC current in a standard four-pole configuration. To measure the current and the voltage, 50-micron golden leads were attached to the surface of the sample in the interface area using silver paint. The temperature dependences of magnetic susceptibility were measured in the temperature range 2–300 K by means of a Quantum Design superconducting quantum interference device magnetometer.

## 3. Results and Discussion

### 3.1. BaTiO_3_/Si Heterostructure

At the first stage bulk structures of silicon, barium titanate in the tetragonal phase and lanthanum manganite with an A type of ferromagnetic ordering (A-AFM) were examined by means of DFT. The explanation for the choice of the phases and magnetic ordering was presented in our previous research [43]. Here, we emphasize that all compounds are semiconductors with indirect band gaps of 0.615 eV, 2.113 eV and 1 eV for Si, BTO and LaMnO_3_, respectively (the unit cells along with the density of states spectra are shown in Figure 1).

For creating the BTO/Si heterostructure, bulk lattice parameters were compared: a = 3.998 Å for BTO and 5.469 Å for silicon. Due to the significant difference in parameters, ε = 27%, to integrate BTO on the silicon substrate, the BTO cell was rotated by 45 degrees around the *z*-axis. Further lattice discrepancy reached ε = 4.2%, which is much more acceptable for modeling the epitaxial growth of BTO on Si. The heterostructures with *n*-layers BaTiO_3_, where *n* = 2–5, were created. The optimized structure with five layers of BTO (only the left half is presented), as well as the calculated density of states spectrum, is demonstrated in Figure 2. In the beginning, for simplification, the calculations were performed without considering the magnetic nature.

It should be noted that the density of the states spectra obtained for the structures with less than five overlayers of BTO look similar, so we do not provide them here. However, each of the considered structures with *n* = 2–5 exhibited the semiconductor-conductor transition. The absence of a band gap for the Si atoms observed in the spectra indicates the presence of conductivity within the silicon substrate as well.

In addition, Figure 2 also shows the presence of structural distortions in the overlayers related to the displacement of titanium atoms out of the oxygen planes. As a result of these displacements, ferroelectric polarization toward the surface of the heterostructure takes place. The dependence of the Ti displacement with respect to the oxygen planes in heterostructures with different numbers of ferroelectric overlayers is presented in Figure 3a. There, “Number of layers” denotes the numbering of TiO_2_ layers; “1” is the interfacial layer. The graph shows the size of displacements in different layers of the slab. In particular, for the green line with five BTO overlayers, one can see that the displacement is the highest in the interfacial layer and it decreases towards the surface. This means that the highest polarization occurs in the interface layers. Secondly, for the heterostructure with the bigger number of layers (ferroelectric film thickness) the displacement of titanium out of the planes is higher within each layer (for instance, compare the first or second point in the figure).

At the next stage, the layer-resolved density of the states spectrum for the heterostructure with five BTO overlayers was examined (Figure 3b demonstrates DOS for Si layers). We identified that, for the Si substrate, the maximum contribution to the conductivity corresponds to the interface atoms (see the interface layer contribution at the Fermi level); then, the DOS at the Fermi level decreases asymptotically to the center of the heterostructure.

The same structure with five layers of ferroelectric was used to analyze the influence of oxygen vacancies (O-vacancies) on the conductive properties of the silicon substrate. The optimized supercell is shown in Figure 4a. The vacancy was placed in the surface TiO_2_ layer of the ferroelectric film, and it is marked by the cross. The layer-resolved density of the states spectrum for the silicon substrate is shown in Figure 4b, demonstrating qualitatively similar behavior to the bare heterostructure, with decreasing DOS at the Fermi level towards the center. However, the difference is that, quantitatively, the DOS contribution at the Fermi level, in the second case with O-vacancy, is slightly lower. That is a surprising feature as the presence of the vacancy provides the additional positive charge and, consequently, the field across the ferroelectric slab should have increased. We propose the following explanation of this effect. In the bare care (as depicted in Figure 2a), all the layers in the ferroelectric slab are neutral in the simple ionic limit: TiO_2_ and BaO. In that case, the out of O planes displacements led to the polarization occurrence towards the interface. In the case of vacancy in the TiO_2_ layer, the resulting charge is (Ti^4+^O^2−^)^2+^, and the final displacement and electrostatic field are lower (see Figure 4a). That fact affects the charge distribution in the Si substrate, as demonstrated in Figure 4c. To summarize, it was obtained that the presence of the vacancy negatively affects the conductivity of the substrate, in particular, the density of states at the Femi-level, decreases (about 20% of the initial state without vacancy).

Lastly, the spin polarized calculations were performed to test the magnetic properties of the BTO/Si heterostructure. In Figure 5, the spin-polarized atom-resolved DOS spectra for the bare 5BTO/Si heterostructure (a) and with oxygen vacancy (b) are presented. The heterostructures used to obtain plots are the same as used above (Figure 2a and Figure 4a, respectively). The spin-polarized calculations revealed that the systems are slightly magnetic; in particular, the system without vacancy is negligibly magnetized, whereas the system with surface vacancy has magnetic moments per surface Ti atoms of 0.183 µ_B_. Thus, the presence of oxygen vacancy does not have a substantial influence on the electronic and magnetic properties. Qualitatively and quantitatively, the behavior does not change significantly. Let us now consider a heterostructure LaMnO_3_/BaTiO_3_ (LMO/BTO) in which, even without the presence of oxygen vacancies, a significant magnetic order can exist at the interface.

### 3.2. BaTiO_3_/LaMnO_3_ Heterostructure

In the previous section, we concluded that the presence of O-vacancies does not change significantly, neither conducting nor magnetic properties of the BTO/Si heterostructure. In order to test the possibilities of a higher magnetization creation in the heterostructures, the system with manganite substrate, i.e.*,* LaMnO_3_/BaTiO_3_, was investigated. The electronic properties of a bare LMO/BTO heterostructure with a varying number of BTO overlayers were discussed in our previous research [35]. Here, we discuss the impact of oxygen vacancies on magnetic properties. In Figure 6, we show that adding the vacancy changes the DOS significantly (Figure 6a–c). That is different from the situation we observed for BTO/Si. Here we observe a semiconductor-conductor transition due to the presence of O-vacancy, located at both the surface and interface. In this case, as shown in Figure 6d,e, the top layer with O-vacancy has changed the charge from (Ba^2+^O^2−^) to (Ba)^+^. Therefore, the out-of-O-plane displacement increases due to the Coulomb attraction of oxygens from the second TiO_2_ layer. In turn, that affects the overall charge distribution along the slab leading to the downshift of states and closing of the band gap.

The magnetic moments distribution here is not affected by the presence of O-vacancy, as in the BTO/Si case. The bare LMO/BTO heterostructure is A-AFM type with ~3.9 µ_B_ per Mn atom and that value changes to ~5 µ_B_ when the vacancy is located in the interfacial MnO layer. Thus, although O-vacancies have a very small effect on the distribution of magnetic moments in LMO/BTO, the magnetization in LMO/BTO is much larger than in the BTO/Si case. However, the overall qualitative picture of the DOS distribution depends on the position of the vacancy. In particular, as seen from comparison of Figure 6b,c the energetical position of the states is similar, but the intensity distribution is different. In addition, the Mn contribution at the Fermi-level is more significant for the case of heterostructure with vacancy located at the interface. That is well understood due to the spatial proximity of Mn and vacancy and corresponding distortions of electronic orbitals. This is similar to the effect discussed for magnetization appearing in LAO/STO due to interface vacancy [12].

To clarify the relationship between the magnetic states and the conducting properties of the interfaces in the LMO/BTO-type heterostructure, magnetic measurements were carried out on such a system in comparison with the results of measurements of the temperature behavior of the resistance.

### 3.3. Magnetic Susceptibility and Resistivity Measurement for Ba_0.8_Sr_0.2_TiO_3_/LaMnO_3_ Heterostructure

The electrical resistance was measured at DC current in a four-pole configuration, as shown in Figure 7a. The X-ray diffraction pattern of the heterostructure is shown in Figure 7b (c = 0.4080 nm for BSTO film). The results of the X-ray measurements indicate that the BSTO film is in the tetragonal phase, which means that this film is in the ferroelectric state. In addition, these results also show that the polarization is directed predominantly in the vertical direction. This is important because it means that the polarization-screening charge is at the interface. Films of solid compositions of the Ba_0.8_Sr_0.2_TiO_3_, depending on the magnitude and sign of deformation, experience ferroelectric phase transitions from the cubic phase to the tetragonal phase with polarization perpendicular to the substrate, or to the orthorhombic phase with polarization parallel to the substrate. In the case of BSTO on LMO, in the configuration when the axis with LMO is perpendicular to the surface, the compressive strain causes the unit cell to shrink in the *a* and *b* directions, and therefore elongate in the ***c*** direction, which effectively stimulates the phase transition to the ferroelectric state. Thus, a rather significant compression strain stimulates the tetragonal phase. Therefore, the BSTO film on the LMO substrates will be in the tetragonal ferroelectric phase up to very low temperatures. Figure 7c shows the temperature dependence of the resistivity for Ba_0.8_Sr_0.2_TiO_3_/LaMnO_3_ (BSTO/LMO) heterostructure. The resistance of the heterostructure decreases with decreasing the temperature and demonstrates a metallic-like behavior below about 160 K. However, the temperature dependence of the resistance for the single LaMnO_3_ crystalline demonstrates the semiconducting behavior of the resistance for any temperature. The temperature dependence of resistivity on inverse temperature is shown in the insert in Figure 7c. The resistance of LMO is shown by blue cycles in Figure 7c. The Ba_0.8_Sr_0.2_TiO_3_/LaMnO_3_ heterostructure shows semiconducting behavior similar to LMO, only above 240 K, when main conduction is carried out through the main volume of the LMO substrate. Below 160 K, the main current flows through the region with a high concentration of carriers with metallic conductivity. At the temperature range 160–240 K, part of the current flows through the interface, reducing the resistance of the heterostructure. The temperature dependence of magnetic susceptibility at a magnetic field of 100 Oe for the Ba_0.8_Sr_0.2_TiO_3_/LaMnO_3_ heterostructure is presented in Figure 7d. Small values of magnetic susceptibility are observed below ~240 K (see insert in Figure 7d), and slightly larger, but still small, values were observed in the temperature range below 140 K. Such values cannot be explained by the transition of the entire volume of the sample to the ferromagnetic state. Therefore, magnetic order occurs only in the small sample volumes: according to the lowest estimates, about 10^−3^ of the total volume of the sample at temperatures below 140 K and less (in about 10^−5^)–at higher temperatures up to ~240 K. Thus, the phase transition to the ferromagnetic phase in the limited volume, probably in the interface area, is observed below 140 K. In addition, ferromagnetic ordering most likely occurs in small uncorrelated areas of the interface, in the temperature range between 140 K and ~240 K. Thus, there is a correlation between the behavior of the ferromagnetic ordering and the behavior of the resistance. Thus, the ferromagnetic ordering of the heterostructure can be associated with the appearance of regions with a high carrier concentration at the interface and the appearance of an exchange interaction through these carriers.

Indeed, the ferromagnetic order in the LaMnO_3_ substrate may appear in the interface area of the BaTiO_3_/LaMnO_3_ heterostructure, and magnetic moments correspond mainly to the Mn atoms [30]. It is known that manganite undergo the transition to a ferromagnetic metallic state upon doping. When La atoms in LaMnO_3_ are substituted by strontium or calcium ions, the samples demonstrate the metallic behavior at temperatures below T_C_ for a high dopant concentration. Ferromagnetism is caused by an indirect ferromagnetic exchange through the current carriers. When free carriers appear at the interface of the Ba_0.8_Sr_0.2_TiO_3_/LaMnO_3_ heterostructure leading to the metallic state, this state will have ferromagnetic properties. This occurs due to a strong ferromagnetic interaction through free current carriers with high concentration at the interface layer. Thus, observed ferromagnetic properties of the interface area have this natural explanation.

## 4. Conclusions

In this work, detailed ab initio calculations of BaTiO_3_/Si heterostructures were presented, and an impact of the oxygen vacancy in the surface layer of the ferroelectric film onto the density of states at the interface was studied. Based on the calculations of the band structures, the possibility of a conducting state in the silicon substrate of the BTO/Si heterostructures was demonstrated. Similar conclusions concerning the electronic states creation at the oxide interfaces with ferroelectrics have been made elsewhere [13,14,15,16]. Here, we show that conductivity may arise in the silicon substrate with intensity decaying away from the interface. In previous works [6,7,8,9,10,11,12], it was demonstrated that vacancies play a crucial role in the electronic and magnetic states of the oxide heterostructures. Here, for the BTO/Si heterostructure, it was shown that oxygen vacancies in the surface layer have a small impact on the density of states at the Fermi level and magnetic moments distribution. In contrast, in the discussed LMO/BTO heterostructure, oxygen vacancies lead to the insulator-conductor transition due to the increase of the electrostatic field across the ferroelectric slab. The reason for that is similar to one observed in the heterostructures of LAO/STO type [6,7,8,9,10,11,12]. In particular, in LAO/STO, conductivity arises due to the charged layers sequence and corresponding electrostatic field across the LAO overlayer, whereas in the heterostructures with ferroelectrics, the field is associated with ferroelectric polarization. It was shown experimentally that ferromagnetic ordering most likely occurs in small uncorrelated areas of the interface in the temperature range between 140 K and ~240 K and in all volumes of the interface below 140 K. There is a correlation between the behavior of the ferromagnetic ordering and the behavior of the resistance. The ferromagnetic ordering at the interface of the heterostructure can be associated with the exchange interaction through current carriers that appear in regions with a high carrier concentration. Thus, the results obtained by DFT for LMO/BTO correlate well with experimental findings for the BSTO/LMO heterostructure.

## Figures and Tables

**Figure 1 materials-15-08334-f001:**
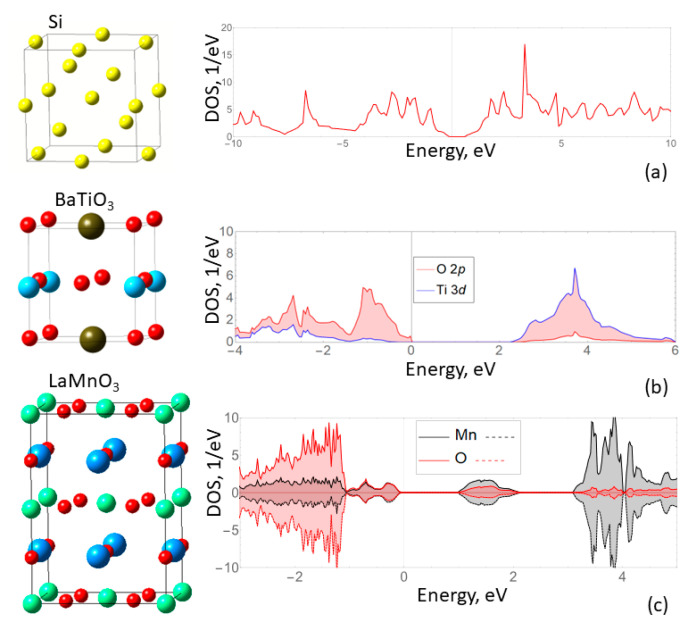
The unit cells of silicon (**a**) barium titanate (**b**) and LaMnO_3_ (**c**), as well as the calculated DOS spectra for these compounds. The DOS spectra confirm the presence of band gaps.

**Figure 2 materials-15-08334-f002:**
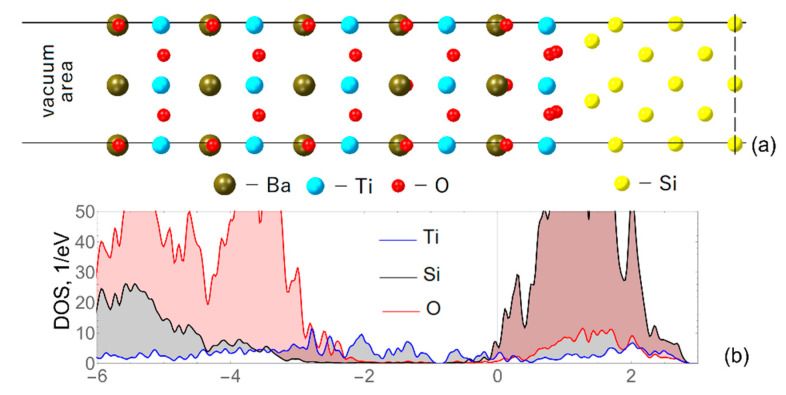
(**a**) The left half of the BaTiO_3_/Si unit supercell obtained after structure optimization. Barium atoms are brown spheres, titanium are blue, oxygen are red, and silicon are yellow. A vacuum area of about 20 Å is also presented, but was not shown in full size. The dashed lane denotes the middle of the supercell. (**b**) The corresponding density of the states spectrum, where the red region is assigned to oxygen atoms, the blue one to silicon atoms, and the black one to titanium atoms. Zero on the energy scale represents the Fermi level. The band gap is completely closed.

**Figure 3 materials-15-08334-f003:**
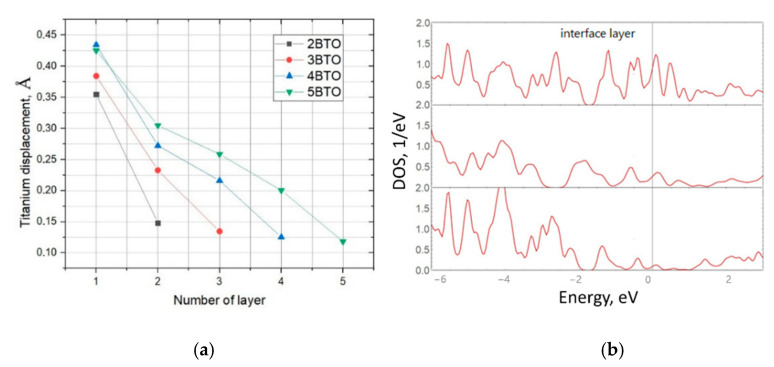
(**a**) The graph shows the dependence of the displacement of titanium atoms with respect to the oxygen plane on the BTO layer number for all obtained structures containing *n*-layers, where *n* = 2–5. The “1” denotes the interfacial layer, whereas “5” is the surface layer of the 5BaTiO_3_/Si heterostructure. (**b**) The layered density-of-states spectrum for a silicon wafer is presented. The first spectrum was compiled for the interface layer; that is, the layer at the silicon-BTO interface, then in turn for the layers to the center of symmetry of the structure.

**Figure 4 materials-15-08334-f004:**
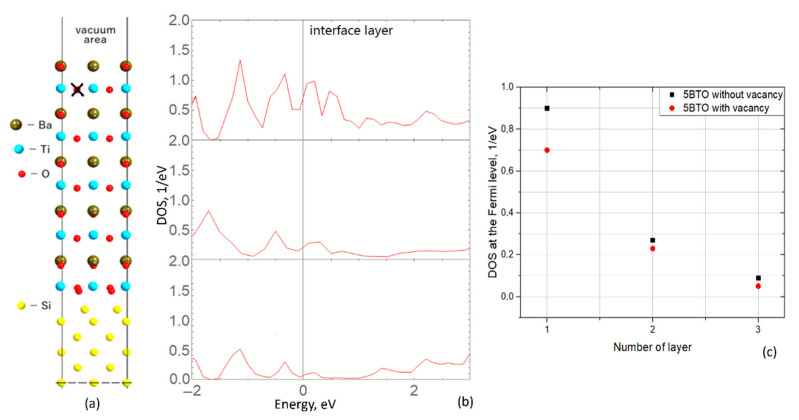
(**a**) The creation of an oxygen vacancy is demonstrated; the structure is identical to that shown in Figure 2. (**b**) Layer-resolved DOS spectra for a silicon substrate of a heterostructure with an oxygen vacancy located at the surface. The upper spectrum is associated with the interface layer; that is, the layer at the silicon-BTO interface, and the bottom one corresponds to the central layer of the modeled substrate. (**c**) The plot of the dependence of DOS values at the Fermi level on the number of layers, where “1” is the interface. A decay towards the center of the substrate is observed here.

**Figure 5 materials-15-08334-f005:**
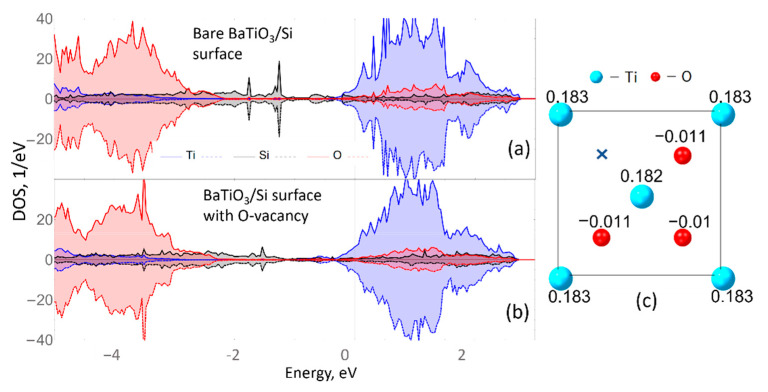
(**a**) Spin-polarized DOS spectra for bare BTO/Si and with (**b**) oxygen vacancy. (**c**) Top view on the surface TiO_2_ layer of the BTO/Si heterostructure with O-vacancy with denoted magnetic moments per atom in µ_B_.

**Figure 6 materials-15-08334-f006:**
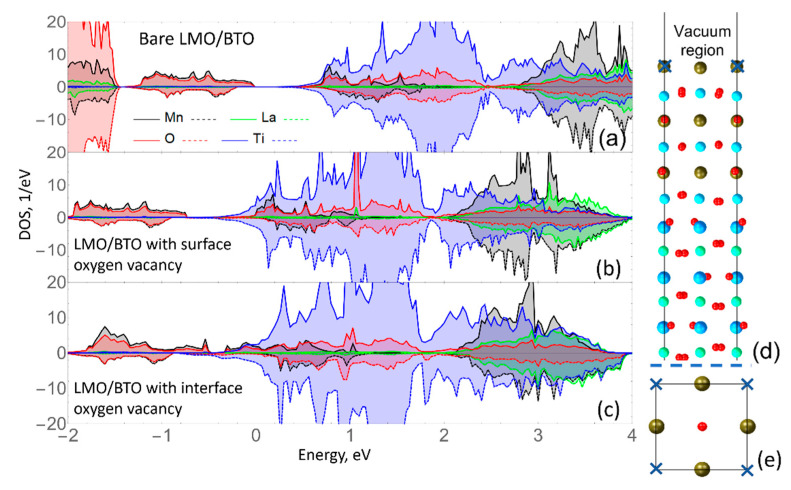
(**a**) Spin-polarized DOS spectra for bare BTO/LMO and (**b**) with oxygen vacancy at the surface and (**c**) at the interface. Side (**d**,**e**) top view onto the LMO/BTO supercell. The upper half is presented here without a full vacuum region.

**Figure 7 materials-15-08334-f007:**
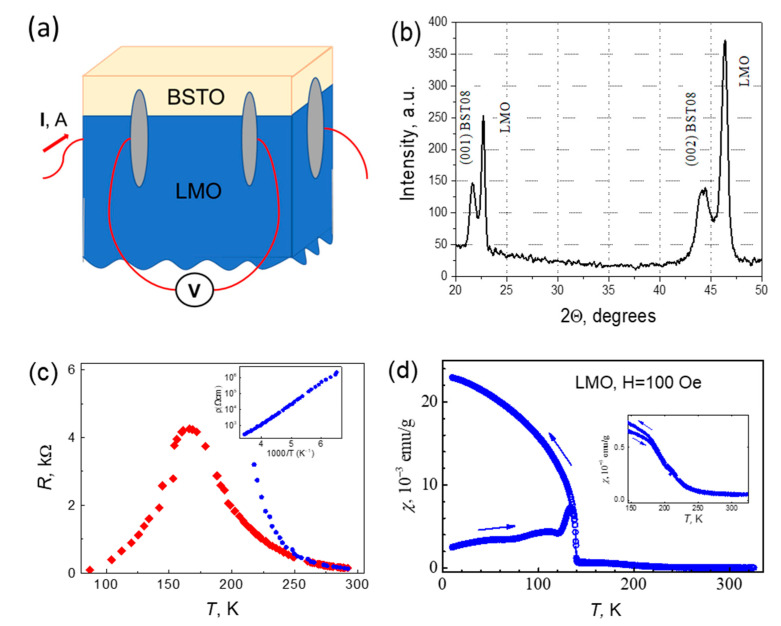
(**a**) The scheme of the electrical resistance measurements with the red color schematically showing the placement of the contacts with grey silver paste and the current and measuring wires; (**b**) the X-ray diffraction pattern of the heterostructure; (**c**) the temperature dependence of the resistivity for the Ba_0.8_Sr_0.2_TiO_3_/LaMnO_3_ heterostructure is shown, the insert shows the temperature dependence of the resistivity for LaMnO_3_ single crystal in the inverse temperature; (**d**) the temperature dependence of magnetic susceptibility for the Ba_0.8_Sr_0.2_TiO_3_/LaMnO_3_ heterostructure. The zoom of the temperature dependence of the magnetic susceptibility in the vicinity of phase transition is shown in the insert.

## Data Availability

The data presented in this study are available on request from the corresponding author.

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
