# Peer review of "DFT Insight into Conductive and Magnetic Properties of Heterostructures with BaTiO3 Overlayer"

_materials, 2022, doi:10.3390/ma15238334_

Round 1
Reviewer 1 Report
The manuscript ID: materials-2029034 “DFT Insight into Conductive and Magnetic Properties of Heterostructures with BaTiO3 Overlayer”. In this article, the author focused on a comparative study of two different structures but with BaTiO3 as an overlayer used as a tool for the property’s manipulation and DFT results for BaTiO3/Si and for BaTiO3/LaMnO3, as well as magnetic susceptibility measurement for Ba0.8Sr0.2TiO3/LaMnO3 heterostructure. The article is sound and well-written, I think this manuscript can be published in this journal with the following minor revisions.
1) The starting point of the introduction is not good/excited. So, the author needs to modify the introduction.
2) At starting, the author needs to write what DFT is. “Density-functional theory (DFT)”.
3) Page No: 3, Figure 2(b): The author needs to assign the curves what are.
4) Please verify each sentence of this paper, Some typo errors.
Author Response
Reviewer #1: The manuscript ID: materials-2029034 “DFT Insight into Conductive and Magnetic Properties of Heterostructures with BaTiO3 Overlayer”. In this article, the author focused on a comparative study of two different structures but with BaTiO3 as an overlayer used as a tool for the property’s manipulation and DFT results for BaTiO3/Si and for BaTiO3/LaMnO3, as well as magnetic susceptibility measurement for Ba0.8Sr0.2TiO3/LaMnO3 heterostructure. The article is sound and well-written, I think this manuscript can be published in this journal with the following minor revisions.
- The starting point of the introduction is not good/excited. So, the author needs to modify the introduction.
The introduction has been modified in accordance with the suggestions.
- At starting, the author needs to write what DFT is. “Density-functional theory (DFT)”.
That correction has been made.
- Page No: 3, Figure 2(b): The author needs to assign the curves what are.
The curves have been assigned.
4) Please verify each sentence of this paper, Some typo errors.
We have checked everything and corrected the whole text.

Reviewer 2 Report
Comments:
1. The novelty/achievements of this work should be highlighted.
2. Some relevant papers in BaTiO3-based heterostructure field have been published in recent years. Some important and latest research results in this field should be mentioned and cited in the section of introduction instead of outdated or earlier papers so that we can offer a solid background and progress to the readers regarding the current state of knowledge on this topic. Therefore, I strongly require the authors to revise this part.
3. Please provide more details on the DFT and Resistivity and magnetic susceptibility measurements details.
4. The authors should make a comparison with the results in recent publications.
5. What about the Band structure change? This characteristic is crucial to evaluate performance. The authors need to add this characteristic and give some comments.
6. The oxygen vacancy is vital for BTO/Si in this manuscript. Therefore, I strongly recommend the authors to add EPR and XPS technologies to investigate oxygen vacancy change and support some views.
7. Microstructure and morphology studies of Ba0.8Sr0.2TiO3/LaMnO3 heterostructure are missed. Thus, XRD and SEM analysis should be performed to better characterize of properties.
8. Finally, there are some grammar mistakes and confusing sentences in the manuscript. The authors need to be corrected.
Author Response
Reviewer #2:
- The novelty/achievements of this work should be highlighted.
The introduction section has been revised along with the conclusions section mentioning the novelty/achievements of this work
- Some relevant papers in BaTiO3-based heterostructure field have been published in recent years. Some important and latest research results in this field should be mentioned and cited in the section of introduction instead of outdated or earlier papers so that we can offer a solid background and progress to the readers regarding the current state of knowledge on this topic. Therefore, I strongly require the authors to revise this part.
Some recent publications with BTO were added along with applications in the Introduction section.
- Please provide more details on the DFT and Resistivity and magnetic susceptibility measurements details.
The main parameters of DFT were presented, and particular details can be found in previously cited papers or by demand from the Authors.
- The authors should make a comparison with the results in recent publications.
All obtained results are in quantitative agreement with previous publications of other research concerning similar problems. Some discussion with papers of other groups has been added in the Conclusion section.
- What about the Band structure change? This characteristic is crucial to evaluate performance. The authors need to add this characteristic and give some comments.
Indeed, band structure can provide significant information, but that was not the purpose of the present research. Besides, the density of states is related to band structure, and both characterize the electronic properties. BS analysis will be done in incoming publications.
- The oxygen vacancy is vital for BTO/Si in this manuscript. Therefore, I strongly recommend the authors to add EPR and XPS technologies to investigate oxygen vacancy change and support some views.
We currently do not have BTO/Si samples for experimental investigation, and therefore experiments on such samples are not discussed. When we have such samples, we will provide comprehensive research jointly with colleagues who work with EPR and XPS technologies.
- Microstructure and morphology studies of Ba0.8Sr0.2TiO3/LaMnO3 heterostructure are missed. Thus, XRD and SEM analysis should be performed to better characterize of properties.
Indeed, all mentioned technics are great tools for structure analysis, but so far that was not the purpose of the present research. That will be done in incoming further publications.
- Finally, there are some grammar mistakes and confusing sentences in the manuscript. The authors need to be corrected.
The corrections were made.

Reviewer 3 Report
Dear authors,
- English and typos should be corrected.
- No discussion with the literature in the results and discussion section. In fact only two previously given reference (without further discussion) are provided.
- Conclusions does not provide the significance of this work at a glance. Needs complete revision.
- Out of 31 references only 6 of them are within the last 2 years. The rest are dated back in decades. This should be definitely improved.
Therefore, i would suggest major revisions, based on the above comments, in order for this manuscript to be able to be considered for publication.
Author Response
Reviewer #3
- English and typos should be corrected.
English checking has been performed
- No discussion with the literature in the results and discussion section. In fact only two previously given reference (without further discussion) are provided.
Some discussion and comparison with previous works were added to the Conclusion section.
- Conclusions does not provide the significance of this work at a glance. Needs complete revision.
The conclusion section has been revised.
- Out of 31 references only 6 of them are within the last 2 years. The rest are dated back in decades. This should be definitely improved.
The publication list has been revised. More recent papers have been added.
Therefore, i would suggest major revisions, based on the above comments, in order for this manuscript to be able to be considered for publication.

Reviewer 4 Report
Please see the attached file.

Author Response
Reviewer #4:
The authors presented an ab initio simulation of ??TiO3 and silicon and monitored the impact of the oxygen vacancy in the surface layer of the ferroelectric film. The reviewer has the following comments/considerations.
- Please add more keywords and define the abbreviation of density functional theory (DFT).
Keywords and the definition of DFT abbreviation have been done in the manuscript.
- Section 2 is given without necessary details; please explain the methodology in detail.
DFT calculation indeed contains lots of details and a huge theoretical background. Within the research we used similar parameters as in our previous publications, here we listed the main of them, which could affect the qualitative results.
- The reviewer did not understand the necessity of the ab initio simulation; please explain.
An ab initio simulation is a comprehensive tool for electronic and magnetic properties prediction. Here, the aim was to find out the common features of two heterostructures with ferroelectric overlayer, and discover the advantages of usage the ferroelectric as a component of the heterostructure.
The sentence concerning that issue has been added at the end of the introduction section.
- Is the silicon lattice doped? If yes, please present the doping concentration.
No, it is a pure Si.
- What about the mobility concentration?
It has not been calculated.
- Please describe the application of this method in FET industry.
The ferroelectric properties of BTO-based heterostructures reignited the interest in the development and implementation of alternative ferroelectric memory concepts, such as ferroelectric field-effect transistors (FFET) [1-4] and negative-capacitance field effect transistors (NCFET) [5, 6].
To date, the FFETs have been demonstrated to exhibit either good retention or endurance, whereas complementary performance characteristics were poor. It is shown that the range of characteristics is quite wide and varies from a few second/hour retention time and endurance up to 10^{12} cycles to a ten year retention time and endurance down to ∼10^3–10^4 cycles [4].
- A. I. Khan, A. Keshavarzi and S. Datta, The future of ferroelectric field-effect transistor technology, Nat. Electron.,
2020, 3, 588–597
- T. Böscke, J. Müller, D. Bräuhaus, U. Schröder and U. Böttger, Ferroelectricity in hafnium oxide: CMOS compatible ferroelectric field effect transistors, in IEEE Int. Electron Devices Meeting, 2011, pp. 24.5.1–24.5.4.
- S. Dünkel, M. Trentzsch, R. Richter, P. Moll, C. Fuchs, O. Gehring, M. Majer, S. Wittek, B. Müller, T. Melde, H. Mulaosmanovic, S. Slesazeck, S. Müller, J. Ocker, M. Noack, D.-A. Löhr, P. Polakowski, J. Müller, T. Mikolajick, J. Höntschel, B. Rice, J. Pellerin and S. Beyer, A FeFET based super-low-power ultra-fast embedded NVM technology for 22 nm FDSOI and beyond. Electron Devices Meeting (IEDM), 2017 IEEE International, 2017, pp. 19.7.1–19.7.4.
- K. Toprasertpong, M. Takenaka and S. Takagi, Direct observation of interface charge behaviors in FeFET by quasi-static split C-V and Hall techniques: revealingFeFET operation, in IEEE International Electron Devices Meeting (IEDM), 2019, pp. 23.7.1–23.7.4. DOI: 10.1109/IEDM19573.2019.8993664.
- W. Cao and K. Banerjee, Is negative capacitance FET a steep-slope logic switch?, Nat. Commun., 2020, 11, 1196.
- M. Hoffmann, F. P. G. Fengler, M. Herzig, T. Mittmann, B. Max, U. Schroeder, R. Negrea, P. Lucian, S. Slesazeck and T. Mikolajick, Unveiling the double-well energy landscape in a ferroelectric layer, Nature, 2019, 565, 464
That was added to the text along with the publication list.
- The English quality of the manuscript should be improved. The reviewer found different grammatical mistakes and typos. Careful proofreading is requested.
English Grammar has been checked and corrected.
- The given references are relatively old published paper. The authors are requested to introduce recently related published papers. The following papers should be suggested in the introduction. • A new method for selective functionalization of silicon nanowire sensors and Bayesian inversion for its parameters https://doi.org/10.1016/j.bios.2019.111527 • Optimal design of nanowire field-effect troponin sensors https://doi.org/10.1016/j.compbiomed.2017.05.008
Suggested papers have been added in the Introduction along with the References list.

Round 2
Reviewer 2 Report
The authors have revised this article based on the reviewer's suggestion. The manuscript can be accepted.
Reviewer 3 Report
Dear authors,
Your comments are well accepted, therefore at this time, i can recommend this manuscript for publication at its current form.
Reviewer 4 Report
The reviewer thanks the authors for denoting the requested comments. The paper is recommended for publication.